# Developing a near Real-Time Cloud Cover Retrieval Algorithm Using Geostationary Satellite Observations for Photovoltaic Plants

Pan Xia [1], Min Min [1,*], Yu Yu [2], Yun Wang [3] and Lu Zhang [4]

[1] Key Laboratory of Tropical Atmosphere-Ocean System, Ministry of Education, and Guangdong Province Key Laboratory for Climate Change and Natural Disaster Studies, School of Atmospheric Sciences, Sun Yat-sen University (Guangdong, Zhuhai), Zhuhai 519082, China

[2] National Meteorological Information Centre, China Meteorological Administration, Beijing 100081, China

[3] China General Nuclear Power Group (CGN), Wind Energy Co., Ltd., Beijing 100106, China

[4] Key Laboratory of Radiometric Calibration and Validation for Environmental Satellites and Innovation Center for FengYun Meteorological Satellite (FYSIC), National Satellite Meteorological Center (National Center for Space Weather), China Meteorological Administration, Beijing 100081, China

* Correspondence: minm5@mail.sysu.edu.cn

**Abstract:** Clouds can block solar radiation from reaching the surface, so timely and effective cloud cover test and forecasting is critical to the operation and economic efficiency of photovoltaic (PV) plants. Traditional cloud cover algorithms based on meteorological satellite observation require many auxiliary data and computing resources, which are hard to implement or transplant for applications at PV plants. In this study, a portable and fast cloud mask algorithm (FCMA) is developed to provide near real-time (NRT) spatial-temporally matched cloud cover products for PV plants. The geostationary satellite imager data from the Advanced Himawari Imager aboard Himawari-8 and the related operational cloud mask algorithm (OCMA) are employed as benchmarks for comparison and validation. Furthermore, the ground-based manually observed cloud cover data at seven quintessential stations at 08:00 and 14:00 BJT (Beijing Time) in 2017 are employed to verify the accuracy of cloud cover data derived from FCMA and OCMA. The results show a high consistency with the ground-based data, and the average correlation coefficient (R) is close to 0.85. Remarkably, the detection accuracy of FCMA is slightly higher than that of OCMA, demonstrating the feasibility of FCMA for providing NRT cloud cover at PV plants.

**Keywords:** cloud cover; photovoltaic plants; geostationary satellite

## 1. Introduction

To reduce global carbon emissions and finally achieve the target of carbon neutralization and carbon peaking, the proportion of renewable energy, such as wind and solar power, in the energy structure of the globe and China will increase significantly in the coming decades [1,2]. Across the globe, the photovoltaic solar energy capacity has increased by approximately 40% per year since 2009. Notably, China is the world leader in the total installed PV capacity and growth rate [3], with approximately 257.1 GW (gigawatts) in August 2021. The installed PV capacity is rapidly increasing, and the value in 2021 was 255.4% more than that in 2016 in China [4–6]. Therefore, it is foreseeable that the PV industry will attract more attention and funding in the future.

Theoretically, the generating efficiency of PV plants primarily depends on the downward solar radiance flux on the solar panels, which is highly susceptible to variations of cloud cover at the PV plant. The scattering and absorption effects of cloud will substantially attenuate downward solar energy, inducing a decrease in generated electric energy. Notably, the considerable uncertainties in the incoming solar energy will cause unpredictable changes in voltage, resulting in the instability of PV power generation. Consequently, high

randomicity and intermittency of PV solar energy induced by cloud cover or movement can have a negative effect on the electric grid system and reduce economic benefits. Hence, accurate and timely monitoring and forecasting of cloud cover or movement is paramount for converting solar energy to electric energy by PV plants [7,8]. In addition, it is also significant and imperative to the transformation of China's energy structure and carbon emission reduction.

Various methods have already been developed and applied to detect instantaneous cloud cover at PV plants. Such a physics-based forecasting method will explicitly consider the effects of cloud cover when it predicts downward solar energy [9]. The physics-based method primarily uses atmospheric and surface parameters retrieved from satellite measurements and other data sources, such as reanalysis data. The cloud cover could be detected and characterized by processing successive sky imageries over PV plants [10,11]. The physics-based method is based on successive imageries from geostationary (GEO) satellites [12,13] or ground-based sky camera [14] measurements, and therefore can predict cloud cover or movement at PV plants up to 6 h in advance. Moreover, common numerical weather prediction (NWP) can also provide an accurate cloud cover forecast from 6 h to the next couple of days in advance. However, the accuracy of these methods is highly dependent on the uncertainties of variables used for forecasting [15]. Moreover, the complicated and labile cloud microphysical properties and the limitations in spatial resolution of satellite imagery and numerical models make it still difficult to accurately predict near real-time (NRT) cloud cover for a long time in advance. It is of great importance for the safe operation, control, and management of PV plants to obtain the ahead cloud fraction, which is also a prerequisite for the grid connection of PV plants [16]. The Advanced Himawari Imager on board the H8 satellite can provide images with much higher spatial, temporal, and spectral resolutions. Thus, we should first accurately determine the cloud cover or fraction at the PV plants. Generally, unique cloud mask or cover algorithms have already been widely developed and implemented for standalone space-based optical sensors, such as the Moderate-resolution Imaging Spectroradiometer (MODIS) [17,18], the Advanced Very-High-Resolution Radiometer (AVHRR) [19], the Advanced Baseline Imager (ABI) on the GEO satellite platform, the Advanced Geostationary Radiation Imager (AGRI), and the Spinning Enhanced Visible and Infrared Imager (SEVIRI) [20].

As mentioned above, cloud mask algorithms based on geostationary satellites can be used to analyze NRT cloud cover. The related algorithms of new-generation geostationary meteorological satellites are very sophisticated. However, the current operational cloud mask algorithm aims to generate some backend science products, weather prediction, and data assimilations [21]. Therefore, it is likely to be unsuitable for obtaining NRT cloud cover at the PV plants. The main problems are as follows: (1) The current cloud mask or cover algorithm does not carefully consider daily variation characteristics of clouds, especially geographical differences at different PV plants, (2) its processing procedure is too complex, costly, and not easy to implement or transplant, and (3) there is not yet a professional cloud cover algorithm for PV stations. For these reasons, the primary objective of this investigation is to develop a portable, fast, and accurate cloud mask algorithm based on GEO satellite data and then provide NRT cloud cover over scatter-distributed in-situ PV plants.

The remainder of this paper is organized as follows. Section 2 introduces the Himawari-8 GEO satellite imager data, the ground-based manually observed cloud cover data, and the mechanisms of operational and fast (or new) cloud mask algorithms. In Section 3, the differences in cloud cover between new and original operational cloud mask algorithms are compared and analyzed. The consistencies between these two algorithms are also analyzed in Section 3. Section 4 summarizes the main conclusions of this investigation.

## 2. Data and Methodology

### 2.1. Geostationary Satellite Imagery Data

Himawari-8 (H8), successfully launched on 7 October 2014, was the new-generation three-axis stabilized operational GEO meteorological satellite of the Japan Meteorological Agency (JMA) (http://www.jma-net.go.jp/msc/en/, accessed on 10 October 2015). Its nadir point on the equator is located at 140.7°E, and the full-disk observation imagery primarily focuses on Japan Island and the Pacific Ocean areas. As the unique optical sensor on board the H8 satellite, the Advanced Himawari Imager (AHI) has 16 earth-view bands from 0.45 μm to 13.3 μm, including three visible (VIS), three near-infrared (NIR), and ten infrared (IR) bands, which routinely execute a full-disk observation mode within a 10-min time interval and fast regional scanning with a 2.5-min maneuver mode. The nominal spatial resolutions of H8/AHI for VIS, NIR, and IR bands are 0.5 km (band at 0.65 μm), 1 km, and 2 km, respectively [22].

The NRT H8/AHI level-1B radiance data with the original horizontal resolution mentioned above only can be downloaded from the China Meteorological Administration (CMA) internal international data exchange File Transfer Protocol (FTP) site. Still, it is sometimes unstable for data transmission. Moreover, we can freely download the H8/AHI Level-1B (L1B) observation radiance data and some Level-2 (L2) science products from the exclusive JAXA (Japan Aerospace Exploration Agency) Himawari data FTP site (ftp.ptree.jaxa.jp) from 7 July 2015 (http://www.jma-net.go.jp/msc/en/, accessed on 10 October 2015) with approximately 2–3 h lag. Our previous studies [23–25] have already illustrated this data acquisition issue for real-time data applications. The unstable and untimely data transmission will negatively impact the monitoring or predicting/extrapolating of cloud cover at PV plants. As one of the general data acquisition methods recommended by JMA, users always use the compact Himawari-8 satellite data-receiving antenna to obtain the NRT down-sampling data from JMA Himawari-Cast in China (see the antenna at the Zhuhai campus of Sun Yat-sen University in Figure 1). To ensure highly efficient data transmission, the down-sampling full-disk H8/AHI data only have 14 bands within 1 km (VIS) and 4 km (NIR and IR bands) horizontal resolutions, lacking 0.47 μm and 0.51 μm bands. However, the timely and high-quality H8/AHI data can still be used to retrieve NRT cloud cover at the PV plants.

### 2.2. Ground-Based Cloud Cover Observation Data

The total cloud cover (TCC) observed by ground-based stations refers to the fraction of the sky covered by all the visible clouds ranging from 0 to 10. It can be used to validate the results of cloud cover from GEO satellite observations. In this study, TCC values were manually observed using the naked human eye three times per day at 00:00, 06:00, and 12:00 UTC, and the observation data were subjected to systematic quality inspection and control. However, vision-obstructing weather, such as blowing snow, sand-dust storms, and fog, may significantly affect the observed TCC results. The visibility observed simultaneously is also categorized into four piecewise ranks (0–2 km, 2–10 km, 10–20 km, and ≥ 20 km) to determine the quality of the observed TCC by the ground-based stations. When the visibility is less than 2 km, the error of manual observation data is large. In this study, these data are regarded as invalid data and are removed. All the visibility data are measured automatically with the visibility meters, and they have been adjusted before they are used [26].

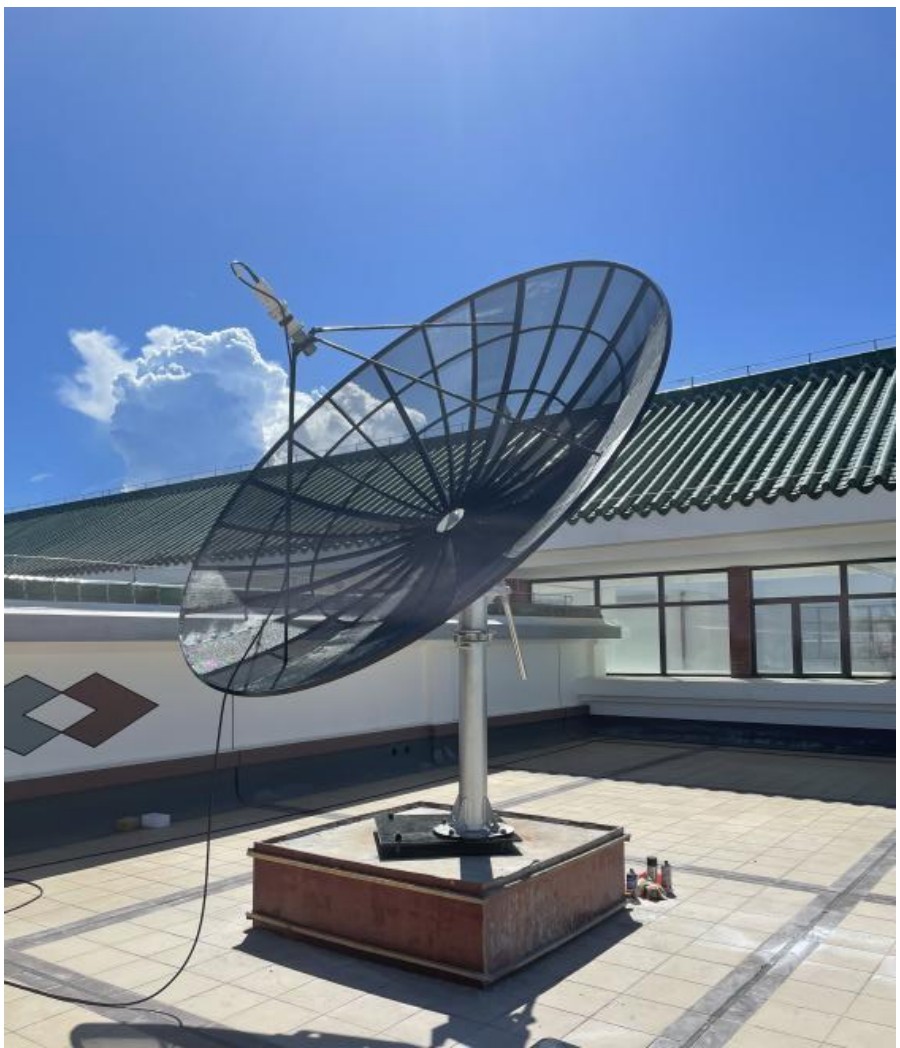

**Figure 1.** Compact near real-time Himawari-8 satellite data receiving antenna on the roof of Haiqin #2 building of Zhuhai campus of Sun Yat-sen University, Guangdong Province, China.

In this investigation, to validate cloud cover retrieved by GEO satellite measurements, we chose seven geographically and climatically representative ground-based stations that are located in different regions of China. The TCC data at these stations at 08:00 and 14:00 BJT (Beijing time) in 2017 are extracted. Figure 2 shows the geographical distributions of these seven selected ground-based stations and the PV plants (http://datasets.wri.org/ dataset/globalpowerplantdatabase, accessed on 1 March 2020) in China. More detailed information on these seven ground-based stations is listed in Table 1. From Figure 2, we find more PV plants are distributed over eastern and northern China, which is attributed to the climatic conditions and the local demands of economic construction for energy. It is noteworthy that the operational TCC manual observations at approximately two-thirds of the ground-based stations in China have already been cancelled by CMA from 1 January 2014. Thus, it is impossible to collect sufficient TCC data to validate the cloud cover in China. Moreover, as introduced before, the main coverage of H8/AHI full-disk is around Japan Island and its surrounding areas, and the regions in western China such as the Tibet Plateau and Xinjiang Province are unable to be aptly observed by the H8/AHI (with the relatively large satellite view zenith angle). Therefore, we chose the Gaolan station located at 103.95°E, 36.35°N as the westernmost station in this study. However, we still resized the down-sampling H8/AHI full-disk L1B data, the horizontal and temporal resolutions of which are 4 km and 10 min, respectively, into a 32 × 32-pixel box to retrieve the cloud mask product [24,27] centered around the seven selected ground-based stations. The observation

range based on the naked eye is approximately 20 km, which is approximately equal to a 5 × 5 neighboring pixel box from a GEO satellite observation or cloud mask product. Therefore, the value of cloud cover rate over ground-based stations using the satellite cloud mask product (*CCRC*) can be defined as follows:

$$CCRC = (a + b)/(5 \times 5) \tag{1}$$

where *a* and *b* are the number of cloudy and likely cloudy pixels [28] in a 5 × 5 neighboring pixel box, respectively (same as the cloud cover from MODIS). As shown in Figure 3d, the station is in the center, and the area enclosed by the red dotted line is the 5 × 5 neighboring pixel box, which is the observation range of the naked eye. The related cloud cover rate from the ground-based station data (*CCRG*) is expressed as follows:

$$CCRG = TCC/10 \tag{2}$$

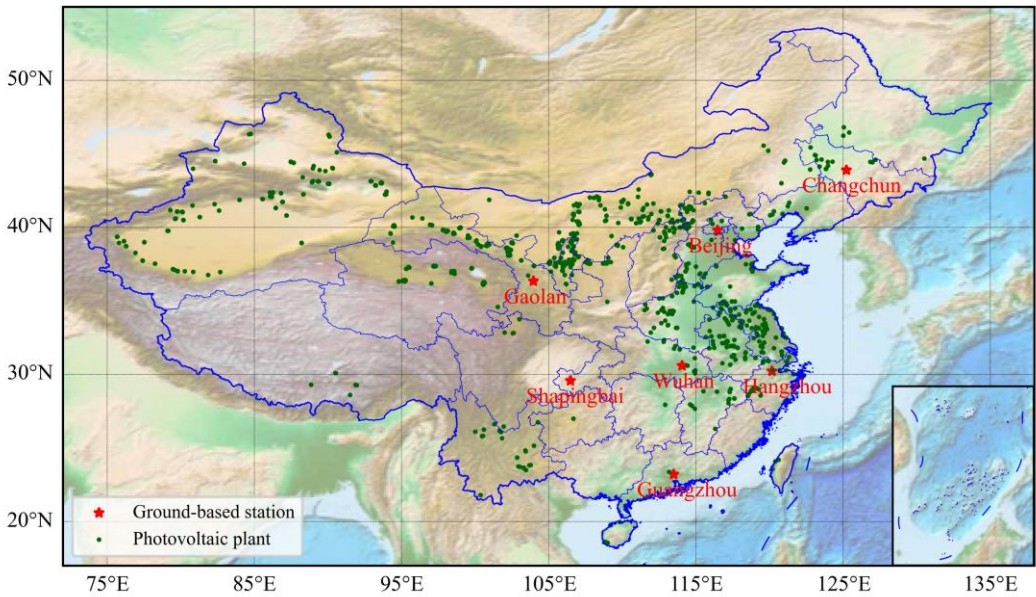

**Figure 2.** Geographic distributions of the PV plants and seven selected ground-based meteorological observation stations in China. The solid green circles and solid red pentagrams represent the PV plants and the ground-based meteorological observation stations, respectively.

**Table 1.** Altitude, surface type, and climate type of the seven selected stations in this study.

| Station | Coordinate | Surface Type | Climate Type | Altitude |
|---|---|---|---|---|
| Gaolan | (103.95°E, 36.35°N) | Valley and basin | Temperate continental | 1520 m |
| Beijing | (116.47°E, 39.80°N) | Plain | Warm temperate semi-humid and semi-arid monsoon | 43.5 m |
| Changchun | (125.22°E, 43.90°N) | Plain | Temperate monsoon | 300 m |
| Wuhan | (114.05°E, 30.60°N) | Hills | Subtropical monsoon | 23.3 m |
| Hangzhou | (120.17°E, 30.23°N) | Hills | Subtropical monsoon | 19 m |
| Shapingbai | (106.47°E, 29.58°N) | Hills and bench terrace | Subtropical monsoon humid | 400 m |
| Guangzhou | (113.48°E, 23.22°N) | Middle and low mountains | Subtropical monsoon | 4.2 m |

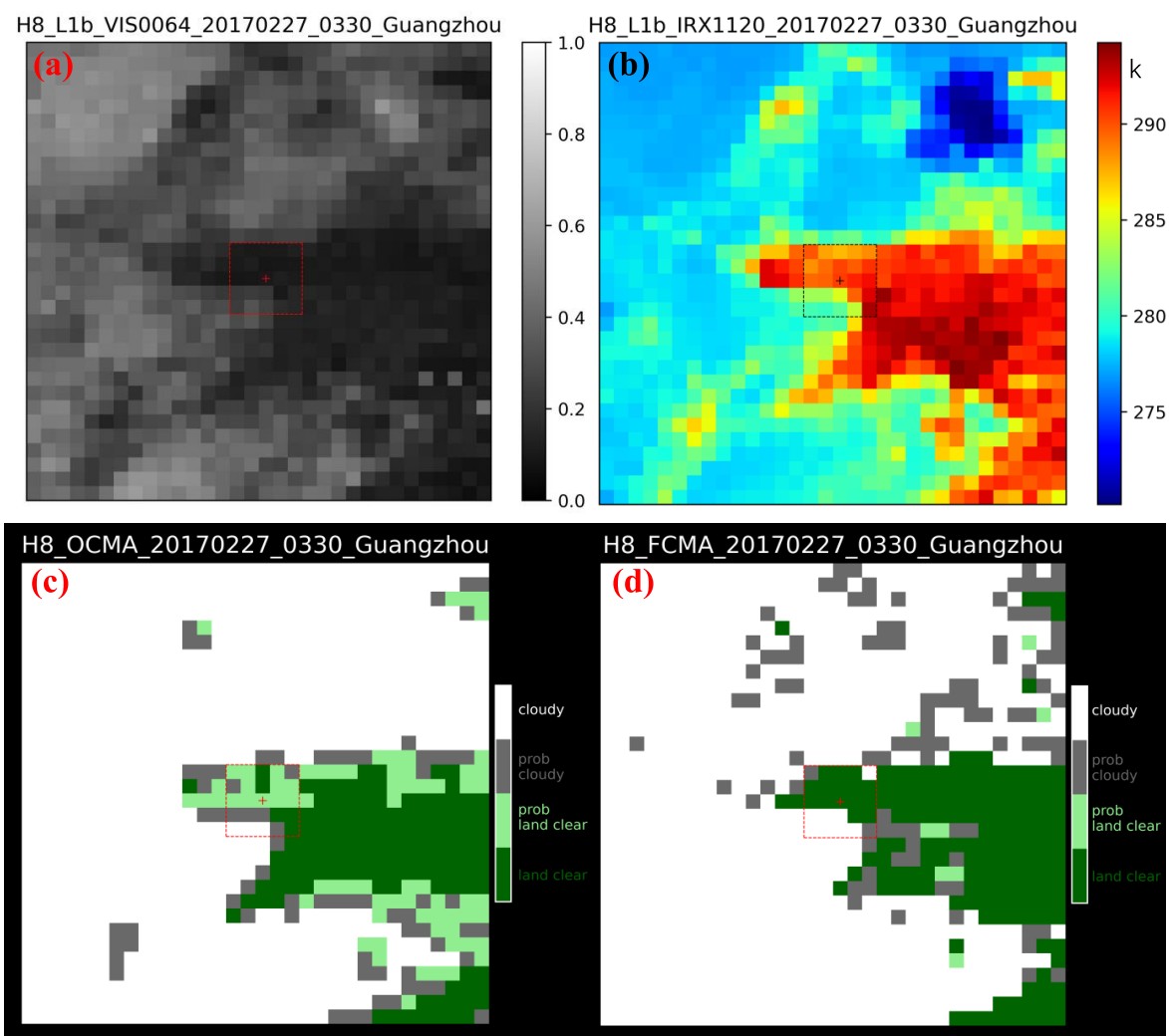

**Figure 3.** Cloud mask results at Guangzhou station retrieved by FCMA (**d**) and OCMA (**c**), and the spatial-temporally matched reflectance (**a**) and brightness temperature (**b**) at 0.64 μm and 11.2 μm bands at 03:30 UTC (11:30 BJT) on 27 February 2017. White, gray, light green, and dark green colors represent cloudy, likely cloudy, likely clear, and clear pixel labels, respectively.

As mentioned above, we should validate the *CCRC* results based on the *CCRG* data in this study.

### 2.3. Multichannel Detection Cloud Mask Data of H8/AHI

Former studies [23–25] have already introduced the operational and unified multi-channel detection cloud mask algorithm for Fengyun-4A (FY-4A) and H8/AHI. It is a four-level cloud mask label product (clear, likely clear, likely cloudy, and cloudy), which is the same as the MODIS product. This robust algorithm primarily includes the IR band test, the shortwave-infrared band test, the solar-reflectance band test, the spatial uniformity test, and the restoral test. These cloudy/clear pixel tests include six infrared tests, two shortwave infrared tests, three solar-reflectance tests, and two spatial uniformity tests. By taking four months of MODIS data as the benchmark, we find the hit rate or accuracy is approximately 91.04% and 91.82% for FY-4A/AGRI and H8/AHI, respectively, indicating the high quality of this operational cloud mask algorithm.

The complex OCMA was initially developed based on the Fengyun Geostationary Algorithm Testbed (FYGAT), which was primarily used for satellite data assimilation and backend science product retrieval [23,29,30]. Although this robust cloud mask algorithm is also applied to calculate the NRT cloud cover at PV plants, its regular operational running

needs many other auxiliary data, such as the numerical weather prediction (NWP) data. Furthermore, it also takes a great deal of time to deal with the spatial-temporally matched NWP data and calculate VIS and IR atmospheric radiative transfer models, which are also used in the backend science product retrieval. Last but not least, the fixed thresholds over land for cloudy/clear pixel tests in this algorithm also do not consider regional and temporal differences, which is likely to introduce some errors in calculating the diurnal cycle of cloud cover [24]. In other words, maintaining such a cloud mask or cover algorithm for scattered PV plants is too expensive and complex. Therefore, it is indispensable to develop a more accessible and more accurate algorithm for monitoring the NRT cloud cover at PV plants by using the data from the antenna receiving device mentioned in Section 2.1.

### 2.4. Fast Cloud Mask Algorithm of H8/AHI for Scattered PV Plants

Considering the cost and efficiency, we have developed a new and fast cloud mask algorithm (FCMA) for scattered PV plants in this investigation, which only works during daytime using six bands of H8/AHI (at 0.64, 0.86, 3.9, 7.0, 11.2, and 12.3 μm) and five inherited and improved cloudy/clear pixel tests from the MODIS official cloud mask algorithm. The five cloudy/clear pixel tests are summarized as follows.

(1)    HVHCT: $H_2O$ Vapor channel ($BT_{7.0\mu m}$) High Cloud Test.

Under the clear-sky situation, the radiance or the corresponding $BT_{7.0\mu m}$ (brightness temperature at the 7.0 μm band calculated by using the classical Planck function) measured by the satellite sensor is emitted by the water vapor in the atmosphere between 200 hPa and 400 hPa. The radiance at the 7.0 μm band emitted by ground or lower clouds will be absorbed by the above atmosphere, which usually makes it undetectable for satellite sensors. Therefore, thick and high clouds above or near the top of the atmospheric layer (approximately 200–400 hPa) will induce a lower-brightness temperature than surrounding pixels that contain clear skies or low clouds. This relatively low BT is caused by the absorption effect of high clouds. On the flip side, it can also be used to test high clouds. The thresholds at the 7.0 μm band for HVHCT in this study are 235 K, 240 K, and 245 K for low confidence, mid-point, and high confidence, respectively.

(2)    BTTCT: $BT_{11\mu m} - BT_{12\mu m}$ Brightness Temperature Thin Cirrus Test.

The difference between $BT_{11\mu m}$ and $BT_{12\mu m}$ or $BTD_{11-12\mu m}$, usually referred to as the split window technique, is widely used for cloud screening of the early AVHRR and GOES (Geo-stationary Operational Environmental Satellite) imager. The $BTD_{11-12\mu m}$ can be used to detect optically thin cirrus clouds because the value of $BTD_{11-12\mu m}$ over thin cirrus clouds is larger than that over clear or likely clear pixels [31]. $BT_{11\mu m}$ and $BT_{12\mu m}$ are usually different, primarily due to the wavelength dependence of optical thickness and the non-linear nature of the Planck function ($B_\lambda$). The $BTD_{11-12\mu m}$ is applied in both OCMA and FCMA algorithms. Note that the BTTCT thresholds are set as a function of the satellite zenith angle, $BT_{11\mu m}$ and $BT_{12\mu m}$.

(3)    BTLCT: $BT_{11\mu m} - BT_{3.9\mu m}$ Brightness Temperature Low Cloud Test.

The difference between $BT_{11\mu m}$ and $BT_{3.9\mu m}$ can also detect the presence of clouds. $BTD_{11-3.9\mu m}$ can effectively detect water clouds in the lower atmosphere in most scenes during the daytime. When the value of $BTD_{11-3.9\mu m}$ becomes much more negative, it is easier to detect non-uniform scenes such as broken clouds. This is consistent with Planck's law that the brightness temperature relies heavily on the warmer portion of the scene and increases with the decreasing wavelength. Since the cloud emissivity at 3.9 μm is prominently lower than that at 11 μm, stratus clouds show positive $BTD_{11-3.9\mu m}$. It is also worth noting that detecting clouds at high latitudes using infrared window radiance data is still a challenge due to very cold surface temperatures. The thresholds for BTLCT in this study are set as −14 K, −12 K, and −10 K for low confidence, mid-point, and high confidence of clear sky, respectively.

(4) VRT: $R_{0.65\mu m}$ Visible Reflectance Test.

VRT is a single-channel threshold cloud test. Theoretically, it is used to distinguish bright clouds from dark surfaces. In this investigation, for PV plants, the water surface type is not considered. The reflectance thresholds at 0.65 μm over land, desert, and snow surfaces where the PV plants are located are developed. These thresholds in the VRT test are the functions of the scattering angle and the background normalized difference vegetation index [32]. The thresholds used in this test are listed in Table 2.

**Table 2.** Thresholds of seven stations used in the VRT test.

| Stations | Threshold [High, Middle, Low] |
|---|---|
| Gaolan | [0.22, 0.18, 0.14] |
| Beijing | [0.22, 0.18, 0.14] |
| Changchun | [0.22, 0.18, 0.14] |
| Wuhan | [0.20, 0.16, 0.12] |
| Hangzhou | [0.22, 0.18, 0.14] |
| Shapingbai | [0.24, 0.20, 0.16] |
| Guangzhou | [0.22, 0.18, 0.14] |

(5) RRT: $R_{0.86\mu m}/R_{0.65\mu m}$ Reflectance Ratio Test.

The principle of the RRT test is that the spectral reflectance at two different shortwave wavelengths (0.86 μm and 0.65 μm) is closer over clouds (the ratio is close to 1) than that of the clear sky over vegetation and water surface. A previous study found that the range of $R_{0.86\mu m}/R_{0.65\mu m}$ is from 0.9 to 1.1 of cloudy pixels by using the AVHRR data. It is worth noting that, for the PV plants distributed in arid and semi-arid areas (Figure 2), the RRT test is very beneficial for cloud detection. The thresholds during the different periods in this test are listed in Table 3.

**Table 3.** Thresholds of the seven stations during different periods in a day used in the RRT test.

| Station | Threshold [Low, Middle, High] | | | |
|---|---|---|---|---|
| | BJT 7:30~09:30 | BJT 09:30~12:30 | BJT 12:30~15:30 | BJT 15:30~16:30 |
| Gaolan | [1.82, 1.87, 1.92] | [1.84, 1.89, 1.94] | [1.81, 1.86, 1.91] | [1.81, 1.86, 1.91] |
| Beijing | [1.84, 1.89, 1.94] | [1.84, 1.89, 1.94] | [1.84, 1.89, 1.94] | [1.82, 1.87, 1.92] |
| Changchun | [1.85, 1.90, 1.95] | [1.89, 1.94, 1.99] | [1.85, 1.90, 1.95] | [1.90, 1.95, 2.00] |
| Wuhan | [1.81, 1.86, 1.91] | [1.83, 1.88, 1.93] | [1.83, 1.88, 1.93] | [1.85, 1.90, 1.95] |
| Hangzhou | [1.86, 1.91, 1.96] | [1.89, 1.94, 1.99] | [1.89, 1.94, 1.99] | [1.90, 1.95, 2.00] |
| Shapingbai | [1.89, 1.94, 1.99] | [1.90, 1.95, 2.00] | [1.90, 1.95, 2.00] | [1.84, 1.89, 1.94] |
| Guangzhou | [1.84, 1.89, 1.94] | [1.88, 1.93, 1.98] | [1.88, 1.93, 1.98] | [1.83, 1.88, 1.93] |

After using five independent cloudy/clear sky pixel tests mentioned above, a general confidence test (GCT) will be conducted. Four different thresholds ($l$, $h$, $p$, $m$) are set here for this confidence test. $l$ is the lower limit of the set, $h$ is the higher limit of the set, $p$ is a power value, and $m$ is the median of the set. Note that the value of $h$-$m$ is equal to $m$-$l$. In the new FCMA algorithm developed in this study, we should iteratively tune the test thresholds for different PV plants. The optimal thresholds determined for each station need a large number of experiments. The confidence value of $c$ is defined as:

$$c = 2^{(p-1)} \times ((I - l)/(2 \times (h - m)))^p \tag{3}$$

After the GCT test, if $c$ is greater than 1, it will be set to 1, and if $c$ is minus, it will be forcibly set to 0. Its dynamic range is from 0 to 1. We use the same four-level label from the classical MODIS cloud mask algorithm here to describe cloudy/clear pixels based on the final test confidence value $c$ ($c > 0.99$ = clear, $0.95 < c \leq 0.99$ = likely clear, $0.66 < c \leq 0.95$ = likely cloudy, and $c \leq 0.66$ = cloudy). Note that the FCMA algorithm only should use the fixed

surface type where the PV plants are located. Therefore, only two surface types of land and desert are considered in this study.

To compare the consistency of the two different cloud mask algorithms mentioned above, the 32 × 32-pixel cloud mask products retrieved by OCMA and FCMA and spatial-temporally matched reflectance and brightness temperatures at 0.64 μm and 11.2 μm bands are shown in Figures 3 and 4. Figure 3 shows the results at Guangzhou station at 03:30 UTC (11:30 BJT) on 27 February 2017 and Figure 4 shows the results at Hangzhou station at 00:50 UTC (08:50 BJT) on 16 August 2017. As can be seen, the consistency between the two cloud mask algorithms is relatively good, which can be confirmed by the visualization results of radiance at the 0.64 μm and 11.2 μm bands. On the other hand, the cloud cover results (the cloud cover values retrieved by OCMA and FCMA in Figure 3 are 0.36 and 0.4, respectively, and those in Figure 4 are 0 and 0.12, respectively) in the 5 × 5 pixels box with a red dotted line agree well with each other. However, differences in the detections of likely land/clear land pixels and likely cloudy/cloudy pixels can still be found between them. The differences between cloudy and likely cloudy pixels, and land and likely land pixels, are primarily attributed to the defining methods used by OCMA and FCMA for these pixels. The spatial uniformity test and restoral test are used in OCMA to identify likely cloudy and clear pixels, which are related to the results of neighbor pixels. However, FCMA uses a final test confidence value of $c$ to distinguish the four categories.

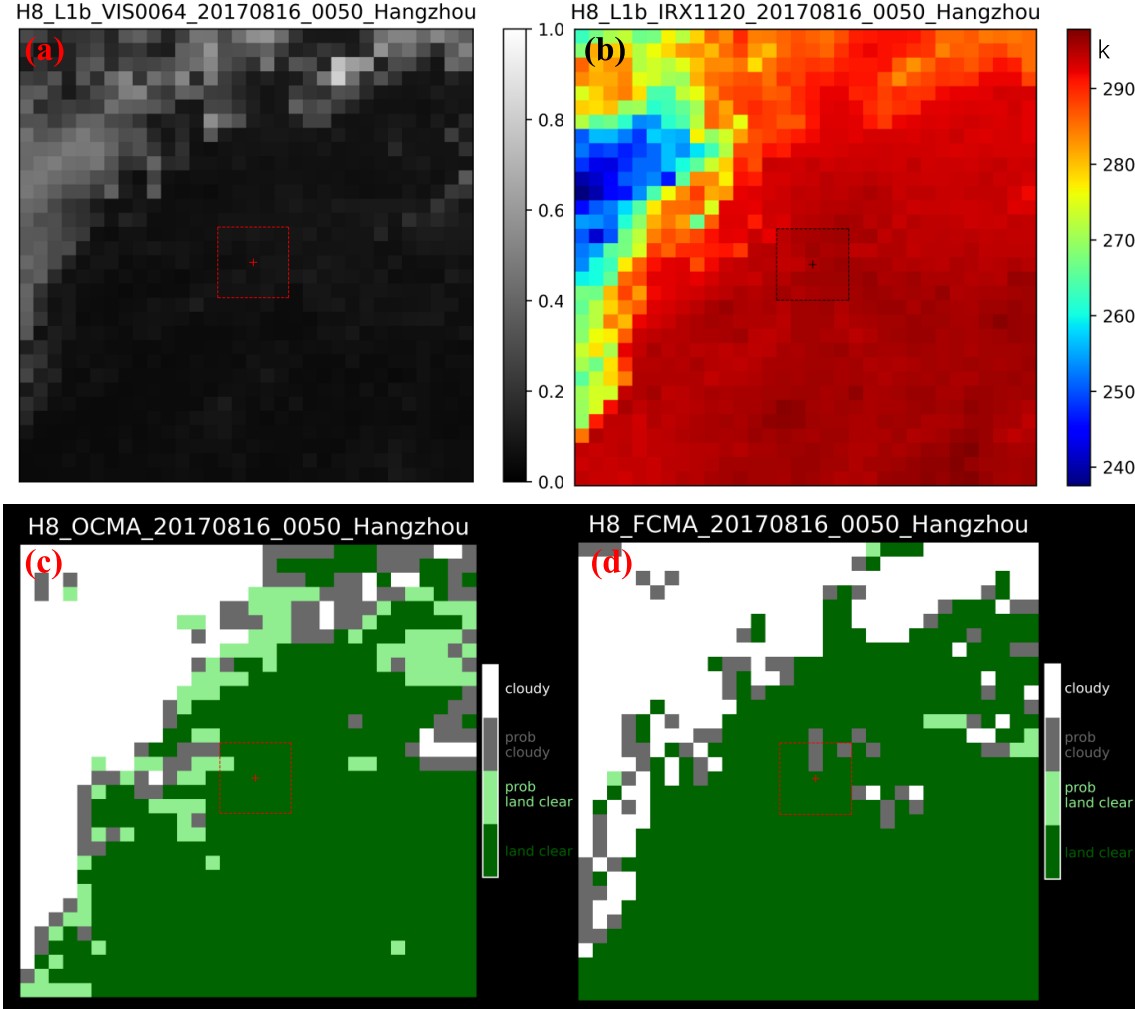

**Figure 4.** Cloud mask results at Hangzhou station retrieved by FCMA (**d**) and OCMA (**c**), and the spatial-temporally matched reflectance (**a**) and brightness temperature (**b**) at 0.64 μm and 11.2 μm bands at 00:50 UTC (08:50 BJT) on 16 August 2017. White, gray, light green, and dark green colors represent cloudy, likely cloudy, likely clear, and clear pixel labels, respectively.



Next, the mean absolute error (*MAE*), root mean square error (*RMSE*), mean bias error (*MBE*), and correlation coefficient (*R*) of cloud cover are employed to evaluate the consistency between two independent algorithms. The results from the OCMA are used as truth here. The definitions of *MAE, RMSE, MBE,* and *R* can be expressed as follows:

$$MAE = \frac{1}{n}\sum_{i=1}^{n}|y_{1,i} - y_{2,i}| \tag{4}$$

$$RMSE = \sqrt{\frac{1}{n}\sum_{i=1}^{n}(y_{1,\,i} - y_{2,\,i})^2} \tag{5}$$

$$MBE = \frac{1}{n}\sum_{i=1}^{n}(y_{1,i} - y_{2,i}) \tag{6}$$

$$R = \frac{\sum_{i=1}^{n}(y_{1,\,i} - \overline{y_1})(y_{2,\,i} - \overline{y_2})}{\sqrt{\sum_{i=1}^{n}(y_{1,\,i} - \overline{y_1})^2}\sqrt{\sum_{i=1}^{n}(y_{2,\,i} - \overline{y_2})^2}} \tag{7}$$

$$\overline{y_1} = \frac{1}{n}\sum_{i=1}^{n}y_{1,i} \tag{8}$$

$$\overline{y_2} = \frac{1}{n}\sum_{i=1}^{n}y_{2,i} \tag{9}$$

where $y_1$ and $y_2$ represent the cloud cover retrieved by the FCMA and OCMA, respectively. $n$ represents the total number of days with effective data.

Three imperative statistical variables ($\Delta Q$, *RMSE*, and *R*) are used in this study to verify the cloud cover from two space-based cloud mask algorithms based on the results from ground-based observation stations. Here, the bias of $\Delta Q$ is defined as follows:

$$\Delta Q = Q_{cld} - Q_{obs} \tag{10}$$

where $Q_{cld}$ is the cloud cover retrieved by FCMA or OCMA and $Q_{obs}$ is the matched manually observed cloud cover from a ground-based station.

## 3. Results and Discussions

### 3.1. Accuracy of Cloud Cover from the FCMA Relative to the OCMA

This investigation evaluates the consistency between the continuous and spatial-temporally matched cloud mask and cover products in 2017 derived from the FCMA and OCMA retrieval algorithms. As known from a previous study [24], after being validated using the spatial-temporally matched MODIS Level-2 cloud mask data, the mean hit rate or accuracy of OCMA is approximately 92%, which illustrates the high quality of OCMA. Figure 5 shows the hourly CCRC results at seven selected ground-based stations from the FCMA and OCMA during the daytime. The hourly data in each box include the six neighboring data before and after the specified hour. For example, the data at 08:00 are composed of the data observed at 07:30, 07:40, 07:50, 08:00, 08:10, and 08:20. As shown in Figure 5a–g, in most cases, the CCRC values of the OCMA are significantly larger than those from the FCMA, indicating the more manifest variation in cloud cover retrieved by the OCMA. However, the average values of these two algorithms are relatively close, and the difference between them is less than 0.1. This finding clearly indicates the excellent consistency of cloud cover between the two algorithms at each hour, especially for the results from 08:30 to 15:30 BJT. It is worth noting that the CCRC results at Shapingbai station retrieved by the FCMA are close to 1, which are remarkably different from those retrieved by the OCMA.

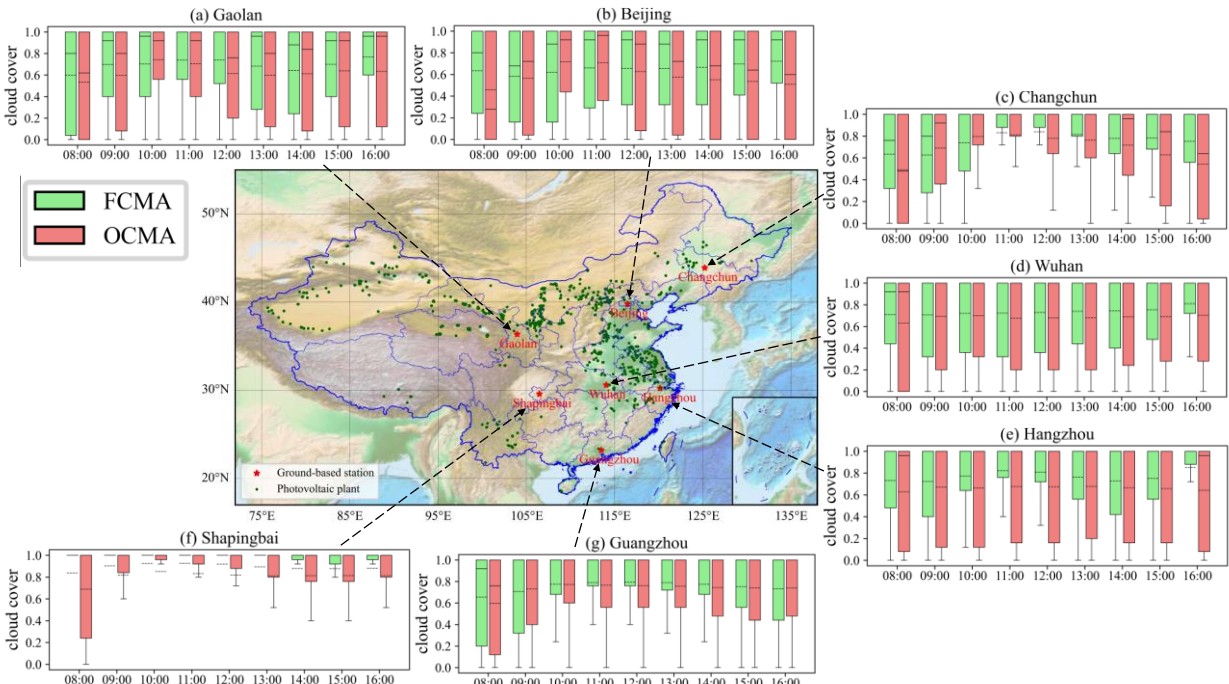

**Figure 5.** Box plots of the hourly mean cloud cover retrieved by FCMA (light green solid box) and OCMA (light coral solid box) during the daytime (from 08:00 to 16:00 BJT) in 2017. The subfigures represent (**a**) Gaolan, (**b**) Beijing, (**c**) Changchun, (**d**) Wuhan, (**e**) Hangzhou, (**f**) Shapingbai, and (**g**) Guangzhou stations. Boxes show the 25th, 50th, and 75th percentiles. The whiskers extend to the most extreme data points between the 75th and 25th percentiles. The dotted line in the box is the mean value.

As in Figure 5, Figure 6 shows the hourly mean results of *MAE*, *RMSE*, *MBE*, and *R* by comparing FCMA with OCMA. Apparently, the results of *MAE*, *RMSE*, *MBE*, and *R* at 08:00 BJT are relatively large, indicating the remarkable difference in the cloud cover values at 08:00 BJT between the two algorithms. The same problem also exists around 16:00 BJT. These differences during the sunrise and sunset times are primarily attributed to the use of visible bands in the FCMA. Before 08:00 BJT and after 16:00 BJT, the instantaneous sun zenith angle is greater than 65°, which will introduce stray light and induce some error in the visible reflectance used in the FCMA.

Furthermore, we also find some differences in the cloud cover results at different stations. For the four stations (Wuhan, Hangzhou, Shapingbai, and Guangzhou) at relatively low latitudes (see Figure 6d–g), the values of *MAE* and *RMSE* are approximately 0.1, the *MBE* is very close to 0, and the *R* exceeds 0.8 in most of the time (8 h) between 08:30 and 16:30, suggesting good results from the FCMA. However, by contrast, Figure 6a–c reveal the relatively large values of *MAE*, *RMSE*, and *MBE* at the three stations (Gaolan, Beijing, and Changchun) with relatively high latitudes. Furthermore, the duration with an *R*-value greater than 0.8 is also substantially shortened to 6 h (from 08:30 to 14:30 BJT). As elucidated above, the thresholds used in the FCMA at each station have already been optimized. The difference at different latitudes is thus primarily attributed to the impact of different satellite and solar zenith angles between these ground-based stations.

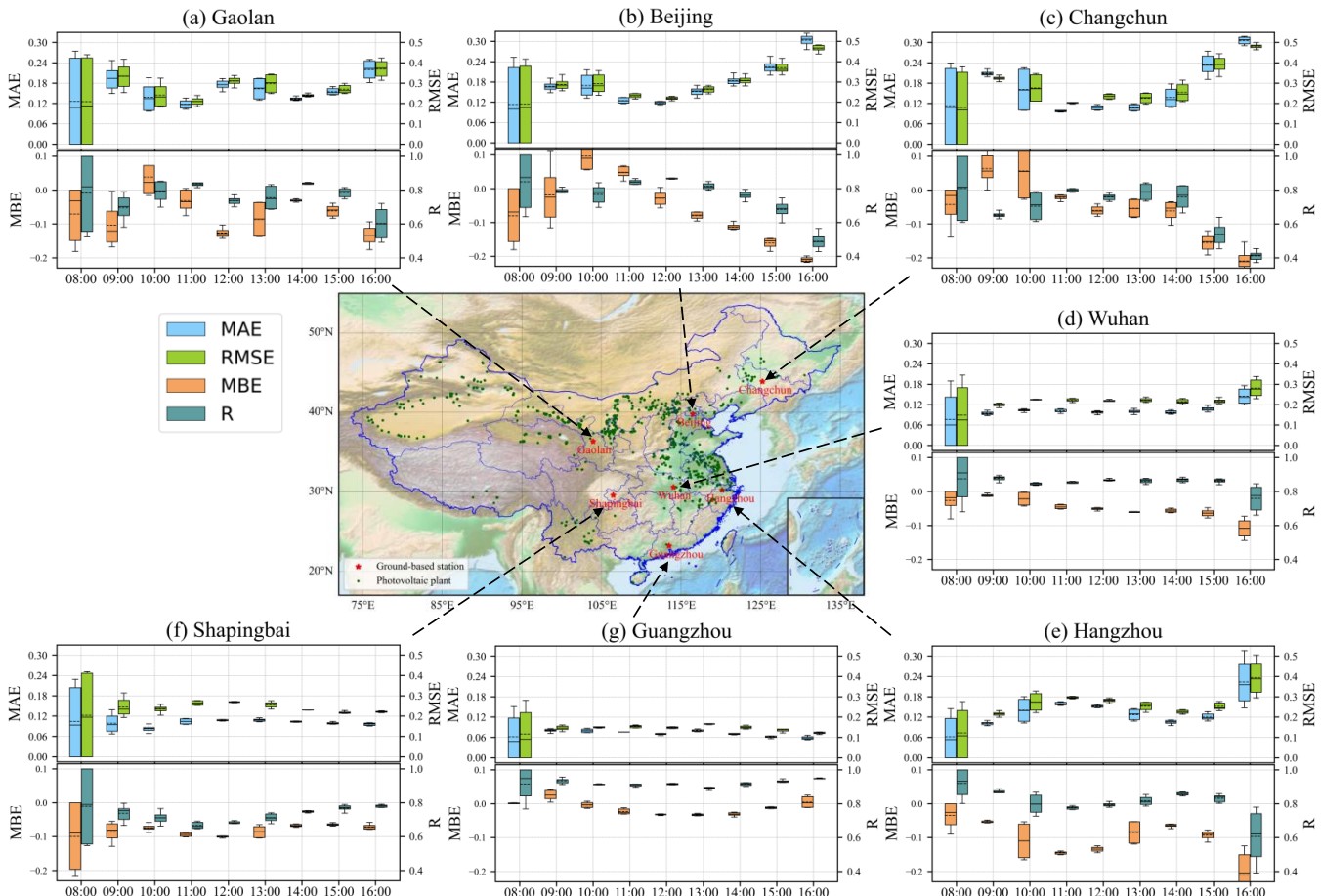

**Figure 6.** Box plots of the hourly mean *MAE* (light sky blue), *RMSE* (yellow green), *MBE* (sandy brown), and *R* (cadet blue) of cloud cover by comparing FCMA with OCMA during the daytime (from 08:00 to 16:00 BJT) in 2017. The subfigures represent (**a**) Gaolan, (**b**) Beijing, (**c**) Changchun, (**d**) Wuhan, (**e**) Hangzhou, (**f**) Shapingbai, and (**g**) Guangzhou stations. Boxes show the 25th, 50th, and 75th percentiles. The whiskers extend to the most extreme data points between the 75th and 25th percentiles. The dotted line in the box is the mean value.

*3.2. Comparisons of Cloud Covers from the FCMA and OCMA with the Ground-Based Observations*

In this section, the cloud cover of seven ground-based stations at 08:00 BJT and 14:00 BJT (during the daytime) in 2017 retrieved by FCMA and OCMA are compared with the manually observed cloud cover data. Before comparison, the cloud cover data from the ground-based station with simultaneous surface visibility of less than 2 km are excluded. This is the result of the avoided error induced in the manually observed data when the ambient visibility is less than 2 km.

Figure 7 shows the inter-comparison of $\Delta Q$ between FCMA and OCMA at 08:00 BJT and 14:00 BJT in 2017. The related *RMSE*, *R,* and sample number of *n* are also listed at the top of the subfigures. For the validation at 08:00 BJT (morning time), the differences in cloud cover between the FCMA and the ground-based stations are not significant from April to October (boreal warm season). Significantly, the duration time for this good consistency with the correlation coefficient exceeding 0.8 and the *RMSE* less than 0.25 is longer at Shapingbai and Guangzhou stations. However, for some stations at relatively high latitudes, such as Gaolan, Beijing, and Changchun, the bias $\Delta Q$ is larger in January, February, March, November, and December (boreal cold season). Sometimes, the value of $\Delta Q$ even exceeds 0.5, indicating the significant overestimation of cloud cover retrieved by FCMA.

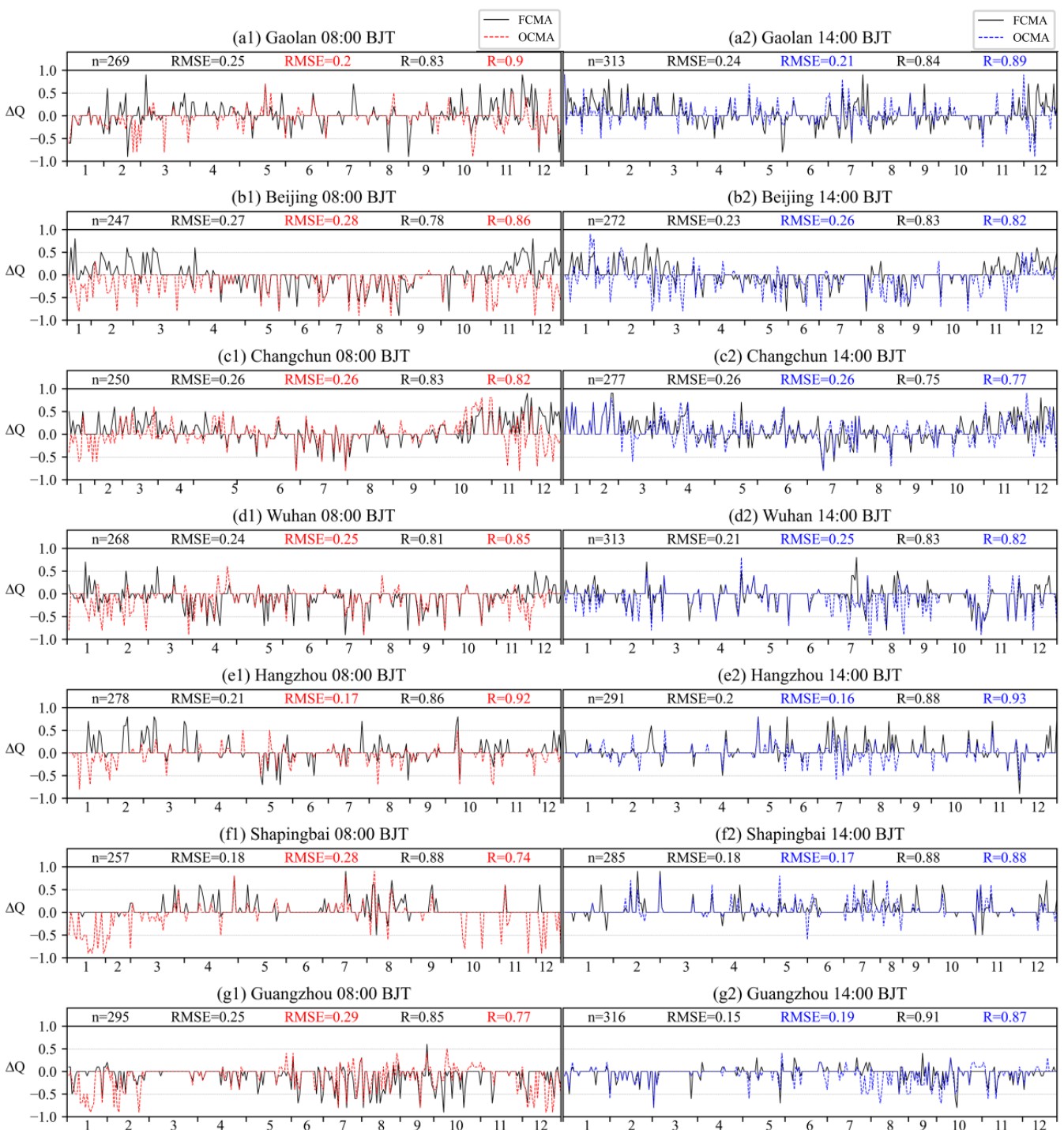

**Figure 7.** Time series of cloud cover biases ($\Delta Q$) of FCMA (solid line) and OCMA (dashed line) at 08:00 BJT (left panel) and 14:00 BJT (right panel) in 2017. The subfigures represent the results at 08:00 BJT for (**a1**) Gaolan, (**b1**) Beijing, (**c1**) Changchun, (**d1**) Wuhan, (**e1**) Hangzhou, (**f1**) Shapingbai, and (**g1**) Guangzhou stations, and at 14:00 BJT for (**a2**) Gaolan, (**b2**) Beijing, (**c2**) Changchun, (**d2**) Wuhan, (**e2**) Hangzhou, (**f2**) Shapingbai, and (**g2**) Guangzhou stations. The corresponding *RMSE* and *R* values are also written at the top of each subfigure. n represents the total number of available days or samples in 2017.

In contrast, the $\Delta Q$ values of OCMA at 08:00 BJT are negative most of the time, implying the significant underestimation of cloud cover retrieved by OCMA. The cloud detection effect of FCMA was very close to OCMA, and even exceeds the OCMA at some

stations, such as Changchun and Guangzhou. Instead, for the validation at 14:00 BJT (afternoon time), the biases between the results of the two algorithms and the true values are manifestly decreasing. It also shows that the values of $|\Delta Q|$ are less than 0.5 most of the time. Despite the considerable improvement at 14:00 BJT, the cloud cover from FCMA and OCMA is still slightly larger and lower than those from ground-based stations, respectively. As seen from the results at 14:00 BJT, more stations show better cloud cover data than the results retrieved by the OCMA.

According to Figure 7, it can be found that the cloud cover accuracies of the two algorithms seem to be remarkably reduced at some stations in the boreal winter season. There are obvious seasonal differences in cloud masks retrieved by these two algorithms, so the data at each ground-based station are divided into four seasons for further analysis. Figure 8 shows the comparisons of correlation coefficient $R$ between the observed cloud cover and the retrieved cloud cover by the two algorithms at the 7 stations at 08:00 BJT and 14:00 BJT in 2017, respectively. The four seasons are divided as follows. Spring includes March, April, and May (MAM), summer includes June, July, and August (JJA), autumn includes September, October, and November (SON), and winter includes December, January, and February (DJF). From the comparison at 08:00 BJT, the cloud cover from these two algorithms is highly correlated with those observed at the ground-based stations in spring, summer, and autumn (close to 0.85). However, in winter, the performance of the OCMA and FCMA decreases. This is likely due to the increase in the solar zenith angle in winter, which reduces the performance in the visible-band-based cloud test. In addition, the visibility in winter is remarkably lower than in other seasons [33], which is prone to cause errors in manually observed cloud cover data. For the results at 14:00 BJT, we can find a relatively high $R$ between them in winter. For other seasons, the $R$ values of the two algorithms are also very consistent with the manually observed data at ground-based stations. However, there are still some sporadically apparent differences in the comparison results at different stations in the same season, such as Gaolan, Wuhan, and Guangzhou stations in summer.

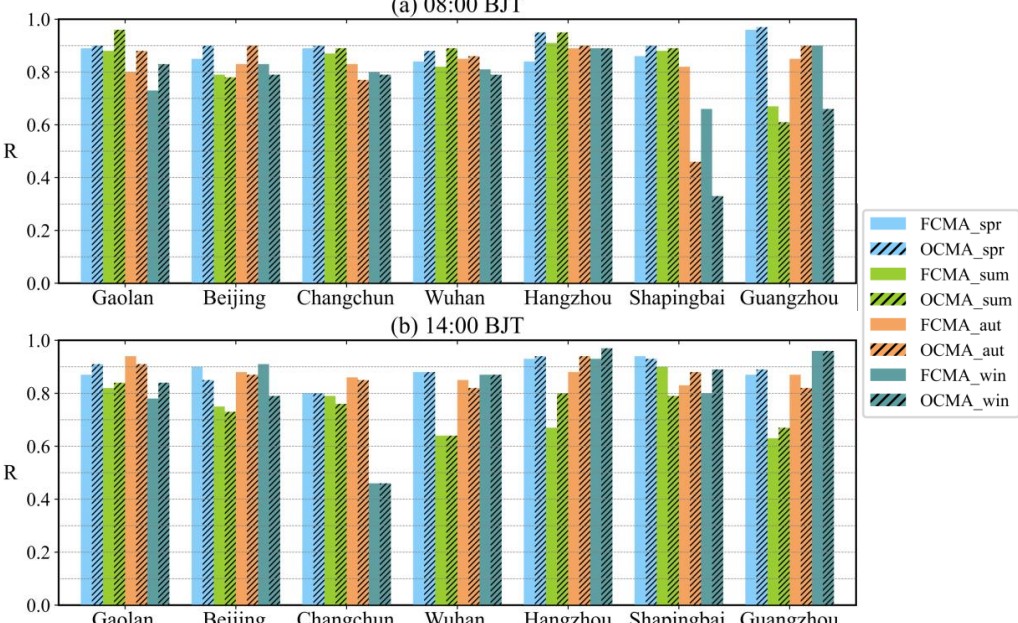

**Figure 8.** Comparisons of $R$ between cloud covers observed by the ground-based stations and retrieved by the two algorithms at seven stations at 08:00 BJT (**a**) and 14:00 BJT (**b**) in 2017. Light sky blue, yellow green, sandy brown, and cadet blue solid boxes represent the seasonal mean results in the boreal spring, summer, autumn, and winter, respectively.

## 4. Conclusions

Based on new-generation GEO satellite data, this study aims to develop a portable, fast, and accurate cloud mask algorithm (FCMA) to provide NRT cloud cover at PV plants. Based on the Level-1B radiance data of H8/AHI, we compared the operational cloud mask algorithm (OCMA) with this new FCMA and verified the accuracy of these two algorithms by using the manually observed cloud cover data at seven quintessential ground-based stations in China in 2017. The main conclusions are summarized as follows.

Through the detection performance of FCMA in Ground-based stations in this paper and considering the diurnal cycle of cloud cover and geographical differences, the fast and new cloud mask algorithm with five independent cloudy/clear pixel tests can retrieve the NRT cloud cover at PV plants. It only uses the level-1B radiance data of the GEO satellite imager without complicated calculations and extra ancillary data.

The FCMA cloud cover data are compared with the OCMA cloud cover data at seven typical stations. The results show that the correlation coefficient $R$ exceeds 0.8, and the $RMSE$ and $MAE$ are approximately 0.13, indicating good consistency between the two independent algorithms. It is also worth noting that the cloud cover results of the FCMA are slightly higher than that of the OCMA (with an average $MBE$ of $-0.1$).

Compared with the manually observed cloud covers at seven typical ground-based stations at 08:00 and 14:00 BJT, the correlation coefficient between the cloud cover of the two algorithms and the ground-based observation is high (the averaged $R$ is close to 0.85). The averaged $R$ of FCMA and OCMA are 0.84714 and 0.82892, respectively. Notably, it also proves that the cloud cover data quality derived from the FCMA is slightly better than that from the OCMA. Moreover, the $RMSE$ at 14:00 BJT is slightly higher than that at 08:00 BJT, which may be related to the effect of the solar zenith angle on visible reflectance.

Overall, the high-quality cloud mask or cover of H8/AHI at PV plants can be retrieved by using the new, portable, and fast FCMA in this investigation. The extensive coverage of GEO satellites facilitates the acquisition of satellite data over each PV plant, which can be used in FCMA to calculate cloud cover. The results aptly demonstrate the reliability of this new algorithm of the GEO satellite imager for retrieving NRT cloud cover products. It is therefore very valuable and less costly to provide accurate NRT cloud cover data for PV plants as an alternative solution. Moreover, it should be noted that researchers using FCMA should be careful when the solar zenith angle is more than $65°$.

**Author Contributions:** Conceptualization, P.X. and M.M.; methodology, M.M.; formal analysis, P.X.; investigation, P.X.; resources, Y.W.; data curation, Y.Y.; writing—original draft preparation, P.X.; writing—review and editing, M.M.; visualization, M.M.; funding acquisition, M.M. and L.Z. All authors have read and agreed to the published version of the manuscript.

**Funding:** This work was supported partly by the Guangdong Major Project of Basic and Applied Basic Research (Grant 2020B0301030004), the National Natural Science Foundation of China (Grants 42175086 and 41975031), and the Guangdong Province Key Laboratory for Climate Change and Natural Disaster Studies (Grant 2020B1212060025).

**Data Availability Statement:** Publicly available datasets were analyzed in this study. These data can be found here: The H8/AHI Level-1B (L1B) observation radiance data and some Level-2 (L2) science products ca be downloaded from the exclusive JAXA (Japan Aerospace Exploration Agency) Himawari data FTP site (ftp.ptree.jaxa.jp) from 7 July 2015 (http://www.jma-net.go.jp/msc/en/, accessed on 10 October 2015) with approximately 2–3 h lag.

**Acknowledgments:** The authors would like to acknowledge the Japan Meteorological Agency for making their Himawari-8 data publicly available. The authors also thank NOAA, NASA, and CIMSS at the University of Wisconsin Madison for freely providing their gfs data, MODIS IMAPP software, and cloud mask algorithm, respectively.

**Conflicts of Interest:** The authors declare no conflict of interest.

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
