# Peer review of "Developing a near Real-Time Cloud Cover Retrieval Algorithm Using Geostationary Satellite Observations for Photovoltaic Plants"

_remotesensing, doi:10.3390/rs15041141_

Round 1
Reviewer 1 Report
In the study by Xia et al. "Deriving near real-time cloud cover at photovoltaic plants using geostationary satellite observations" a new cloud detection algorithm is developed based on Himawari-8 satellite data aiming at a near real-time cloud cover product for PV plants without additional data. The study set-up is done in a structured and concise way to compare and validate the new cloud masks with an existing algorithm and ground based observations, respectively. The study concept and idea fits to the scope of the journal of Remote Sensing and results will have implications for other low-cost cloud masks algorithms.
Minor deficits are seen in the main message presented in the abstract and conclusion of the manuscript and stating that the new algorithm is better than the operational cloud mask algorithm. Here, limitations of the new cloud mask algorithm should be mentioned as well.
Minor comments:
"These cloudy/clean pixel tests include six infrared tests, two shortwave infrared tests, three solar-reflectance tests and two spatial uniformity tests (see Figure 1)."
Why is this sentence referring to Figure 1?
"H08/AHI" vs. "H8/AHI".
Please use a consistent abbreviation throughout the manuscript.
"(1) HVHCT: H2O Vaper channel (BT 7.0μm ) High Cloud Test."
--> Vapor
Table 2:
Is "[low, middle, high]" referring to the confidence level? Please specify.
"clean sky"
--> clear sky. Please be consistent throughout the manuscript.
"It is worth noting that the CCRC results at Chongqing station retrieved by the FCMA are close to 1"
The station Chongqing is not listed anywhere else in the tables or figures.
"There are obvious seasonal differences in cloud mask retrieved by these two algorithms"
Did you mean "seasonal differences in accuracy of the cloud masks"?
Conclusion
"The results show that the correlation coefficient R exceeds 0.8".
This is only true for selected stations and time slots.
“can retrieve the NRT cloud cover at PV plants.”
It is not clear how the locations of the PV plants are used in this analysis. Is the data just used for visualisation or do you assume that if the algorithm is verified at the ground-based stations it will also perform at the locations of the PV plants? Please add this information.
How is the performance of these algorithms affected over arid regions such as deserts? Low cloud detection is usually very difficult over such bright surfaces.
"Notably, it also proves that the cloud cover data quality derived from the FCMA is slightly
better than that from the OCMA."
On which number this statement is based? Please indicate if this refers to selected stations/time slots or an average number.
Author Response
thanks for your suggestions. More responses to comment can be found in the attached file.

Reviewer 2 Report
Attachment

Author Response

(The authors gave the same response as above.)
